# Towards Group-aware Search Success

## Abstract

Traditional measures of search success often overlook the varying information needs of different demographic groups. To address this gap, we introduce a novel metric, named Group-aware Search Success (GA-SS). GA-SS redefines search success to ensure that all demographic groups achieve satisfaction from search outcomes. We introduce a comprehensive mathematical framework to calculate GA-SS, incorporating both static and stochastic ranking policies and integrating user browsing models for a more accurate assessment. In addition, we have proposed Group-aware Most Popular Completion (gMPC) ranking model to account for demographic variances in user intent, aligning more closely with the diverse needs of all user groups. We empirically validate our metric and approach with two real-world datasets: one focusing on query auto-completion and the other on movie recommendations, where the results highlight the impact of stochasticity and the complex interplay among various search success metrics. Our findings advocate for a more inclusive approach in measuring search success, as well as inspiring future investigations into the quality of service of search.

## 1 Introduction

Search is one of the primary methods for people to fulfill their information needs. Typically, users input a query $q$ to express their information needs and intents $t$, prompting search systems to return a list of ranked items $d$. Transitioning from the mechanics of query input to outcome assessment, measuring search success becomes pivotal. The most intuitive method to gauge the success of a system is by averaging the satisfaction of all individuals. We argue that this is suboptimal and cannot distinguish certain nuances. Consider two equal-sized searcher groups $g_A$ and $g_B$ and a query $q$ that corresponds to two equally dominant intents $t_1$ and $t_2$ (see Case 1 in Figure 2). For simplicity, we assume group $g_A$ is only interested in $t_1$ and group $g_B$ only interested in $t_2$. If a search system retrieves 50% of items that are relevant to each intent, it would appear equally successful as one that exclusively retrieves items relevant to $t_1$. However, the latter scenario leaves group $g_B$ entirely unsatisfied, highlighting a situation that traditional search success measurement may fail to distinguish.

Recent studies have expanded the definition of search success by introducing criteria such as diversity [1, 6]. It might appear that optimizing towards diversity could mitigate the aforementioned limitations, especially since a system that retrieves items relevant to multiple intents naturally seems more diverse—and consequently more successful—than one focused on a single intent. However, introducing diversity alone does not capture all the nuances involved. Consider a scenario where a query $q$ aligns with four dominant intents: $t_1$, $t_2$, $t_3$, and $t_4$ (see Case 2 in Figure 1). Suppose group $g_A$ is equally interested in $t_1$, $t_2$, and $t_3$, while group $g_B$ focuses solely on $t_4$. In this case, a system that predominantly retrieves items

*Conference acronym 'XX, June 03–05, 2018, Woodstock, NY*
2022. ACM ISBN 978-1-4503-XXXX-X/18/06...$15.00
https://doi.org/XXXXXXX.XXXXXXX

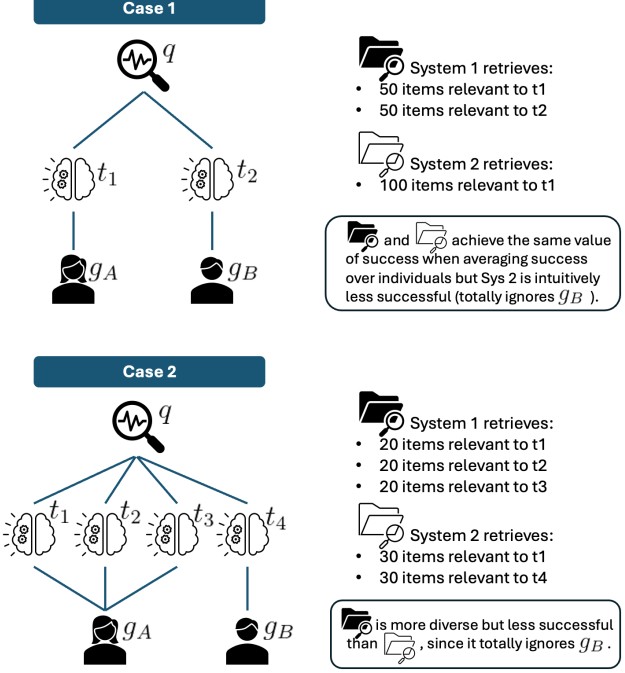

**Figure 1: Two motivation examples to show that previous search success measures cannot distinguish certain nuances. Each edge in the figure between the query ($q$) and intent ($t$) carries equal weight, signifying that the query is uniformly relevant to the connected intents. Similarly, the edges linking the user group ($g$) to the intent ($t$) have equal weight within each group, indicating that members of the group have a uniform level of interest in the associated intent.**

relevant to $t_1$, $t_2$, and $t_3$ would appear more diverse compared to the other system that effectively balances the retrieval between $t_1$ and $t_4$. However, the former one is less successful since group $g_B$ is totally ignore. This demonstrates the limitations of using diversity as the sole measure of search success.

Inspired by the shortcomings identified in previous search success measures, we advocate for a refined and nuanced definition of search success that accounts for the impact across diverse demographic groups. The crucial insight is that traditional metrics often overlook the varied intent distributions that different groups may have towards the same query. In this work, we introduce the concept of Group-aware Search Success (GA-SS), filling a gap in existing literature. We define search success as being achieved *iff* all demographic groups find success in their search outcomes. Let $g \in G$ represent a group of searchers, $t \in \tau$ an information need or intent, $q \in Q$ a query, and $\sigma_q$ the ranked list of items or documents retrieved in response to $q$. We define $s$ as the event of a successful search. The probability of achieving group-aware search success

(GA-SS), $p(\mathrm{S}^{\mathrm{GA}}|q)$, for a given query $q$ is the joint probability that all groups are successful, mathematically defined as:

$$p(\mathrm{S}^{\mathrm{GA}}|q) = \prod_g p(s|q, g). \tag{1}$$

Building on our proposed Group-aware Search Success (GA-SS) metric, the rest of this paper shows a detailed derivation of the metric, explaining how to decompose it and apply either static or stochastic ranking policies, alongside incorporating a user browsing model for computation. We then study the metric relations for within-query and across-queries using a multi-query toy example to demonstrate that enhancements in one metric do not always correlate and may, in fact, negatively affect the other. Subsequently, we employ two real-world datasets, one for query auto-completion and the other for movie recommendations, to empirically study the impact of stochasticity and the correlations among various search success metrics, providing robust support for our initial observations from illustrative examples. Moreover, we enhance the traditional Most Popular Completion (MPC) ranking model to include a group-aware approach, taking into account the varying interests of different demographic groups. Additional case studies on movie recommendations illustrate the efficacy of our method and the impact of stochasticity, highlighting a crucial trade-off between fairness and success in real-world search systems. Finally, we show the connection of our proposed search success to relevance metrics for quality of service in search and provide insightful future research directions. In summary, our main contributions are:

- We propose and formulate the Group-aware Search Success (GA-SS) and introduce a group-aware adaptation of the Most Popular Completion (MPC) ranking model. This represents a novel framework that quantifies search success by incorporating the diverse intents of different demographic groups.
- We provide a detailed theoretical derivation and methodological advancements in the computation of GA-SS, employing both static and stochastic ranking policies with some user browsing model.
- We conduct detailed analysis using two real-world datasets to study the impact of stochasticity and the correlations among various search success metrics, offering practical insights into search success measurement.

## 2 Related Work

In this section, we review related works from the following three topics: (i) diversity in search, (ii) fairness in search, and (iii) ranking with stochastic policy.

### 2.1 Diversity in Search

Since the late 20th century, diversity in search has garnered significant attention, starting with the introduction of the Maximal Marginal Relevance (MMR) method by Carbonell and Goldstein [5]. Building on this, Clarke et al. [6] developed a framework that systematically rewards novelty and diversity for search systems, leading to numerous studies aimed at enhancing these aspects.

Radlinski et al. [25] identified two primary categories of search diversity: extrinsic diversity and intrinsic diversity. Extrinsic diversity deals with the uncertainties in search queries, which can arise from either ambiguity or variability in user intent. For instance, the query *"jaguar"* may refer to different concepts, and a query like *"BioNTech, Pfizer vaccine"* can elicit varied information needs from different users, such as patients, doctors, or entrepreneurs [38]. This diversity type aims to provide comprehensive search results that cater to these varied interpretations and needs. On the other hand, intrinsic diversity focuses on reducing redundancy within the search results themselves, even for queries with a clear and single intent. This approach enhances the novelty of the results, as seen in a query for *"jaguar as an animal"*, where diverse images of jaguars from various angles are preferred over repetitive views. The distinction between extrinsic and intrinsic diversity lies in their approaches to enhancing user satisfaction: extrinsic diversity addresses multiple interpretations of a query, while intrinsic diversity enriches the content quality for specific intents. Both are essential for a search system to meet a wide range of user needs while keeping the content fresh and engaging.

Our introduction of group-aware search success shows relation to the diversity foundation by specifically tailoring search results not just to individual users, but to groups of users. It acknowledges that different groups may require varying distribution of intents, thereby extending the concept of diversity from individual queries to collective interactions.

### 2.2 Fairness in Search

Fairness in search and recommendation has received increasing attention in the community. There is no single agreed-upon definition of fairness, but it commonly involves considering the perspectives of various stakeholders, such as: (i) consumers (i.e., users seeking content), (ii) producers (i.e., content creators or publishers), (iii) information subjects (i.e., individuals featured in search items, such as job candidates), and (iv) other side stakeholders (i.e., groups not directly using the system but affected by it).

Addressing fairness requires evaluating multiple dimensions for each stakeholder group. For consumers, this includes ensuring equitable quality of service across demographics. Ekstrand et al. [10] and Neophytou et al. [20] have shown variability in recommender system performance across different demographic groups. Similarly, Mehrotra et al. [18] observed disparities in web search contexts. Wu et al. [36] tackle this by modeling fairness in service quality as variations in Normalized Discounted Cumulative Gain (NDCG) among user groups, seeking to minimize these differences during model optimization. Wu et al. [37] later propose novel fairness notions to consider group attributes for multi-sided stakeholders to identify and mitigate fairness concerns that go beyond individuals in search and recommendation.

Fairness also pertains to the content exposure received by consumers. Disparate exposure to economic opportunities, for instance, can lead to allocative harms. In job search systems, it is crucial to balance exposure to different job levels across demographics [4]. Previous research has explored notions of social fairness such as Envy-freeness [23] and Least Misery [14] to address fair allocation. Furthermore, exposure disparities can perpetuate consumer stereotypes, as observed in news recommenders that might reflect gender-based biases [35]. Techniques using adversarial learning and domain-confusion [7, 32] have been employed to develop representations that obscure protected attributes like race or gender,

as explored in studies by Zhang et al. [40], Bose and Hamilton [2], and Rekabsaz et al. [29].

Our group-aware search success shows some relation to fairness in search. By considering group dynamics, we aim to ensure equitable information access and success towards various searcher groups. Our concept also concerns item exposure during our modeling process which will be later detailed.

## 2.3 Ranking with Stochastic Policy

Early works in learning-to-rank for IR [15] mostly focus on static ranking policies that produce static ordering of items given a user launched query. Inspired by Pandey et al. [22], who initially suggested incorporating randomization into ranking systems, numerous studies have utilized randomization to gather unbiased implicit feedback from user behavior data [12, 21, 26, 27, 33]. This strategy also aids in training unbiased ranking models using biased user feedback [11]. Moreover, stochastic ranking policies have been used to enhance the diversity of search results [28] and to promote fairer exposure of information content [9, 31, 39]. Building on these applications, recent studies [3, 9, 21, 31, 39] have focused on optimizing stochastic ranking policies to achieve these goals. In Section 3, we will demonstrate how stochastic ranking policies can be employed towards our proposed Group-aware Search Success.

## 3 Group-aware Search Success

As introduced in Section 1, existing measures of search success often overlook the searcher group information thus are insufficient to represent a search system's real success. To address this, we introduce a theoretical framework that defines Group-aware Search Success (GA-SS). Our definition takes into account the impact of varying intent distributions associated with queries from different demographic groups.

## 3.1 Group-aware Search Success within Query

To achieve a high quality of service for a given query, we define search success *if and only if* all groups are successful as shown in Eq. 1: $p(S^{GA}|q) = \prod_g p(s|q, g)$. To further compute the probability of success given a query and group, $p(s|q, g)$, we need to be aware that the intent of different user groups searching for the same query may not be the same. Following Agrawal et al. [1], we use an intent-aware setting and further have:

$$p(s|q, g) = \sum_t^\tau p(t|q, g) \cdot p(s|t, q, g), \qquad (2)$$

$$= \sum_t^\tau p(t|q, g) \cdot p(s|t, q). \quad \text{if } s \perp\!\!\!\perp g \mid t, q \qquad (3)$$

Based on this framing, the GA-SS within a given query can be finally written as below:

$$p(S^{GA}|q) = \prod_g p(s|q, g), \qquad (4)$$

$$= \prod_g \left( \sum_t^\tau p(t|q, g) \cdot p(s|t, q) \right). \qquad (5)$$

## 3.2 Group-aware Search Success across Queries

To consider the success of the entire system, we need to define search success across queries through aggregation over Eq. 1. This aggregation has two dimensions: (i) success across various queries, and (ii) success across different demographic groups. Depending on the order of aggregation, we can define GA-SS in the following two distinct ways.

- $\sum_q \prod_g$: a search system is successful if it is successful for all queries, where for each query it should be successful over all demographic groups.

$$p(S^{GA}_{\sum \prod}) = \sum_q p(S, q), \qquad (6)$$

$$= \sum_q p(S|q) \cdot p(q), \quad q \text{ has no overlap} \qquad (7)$$

$$= \sum_q \left( \prod_g p(s|q, g) \right) \cdot p(q), \qquad (8)$$

$$= \sum_q \left( \prod_g \left( \sum_t p(t|q, g) \cdot p(s|t, q, g) \right) \right) \cdot p(q), \qquad (9)$$

$$= \sum_q \left( \prod_g \left( \sum_t p(t|q, g) \cdot p(s|t, q) \right) \right) \cdot p(q). \quad \text{if } s \perp\!\!\!\perp g \mid t, q \qquad (10)$$

- $\prod_g \sum_q$: a search system is successful if it is successful over all demographic groups, where the success for each group is based on the success across all queries.

$$p(S^{GA}_{\prod \sum}) = \prod_g p(s|g), \qquad (11)$$

$$= \prod_g \left( \sum_q p(s, q|g) \right), \qquad (12)$$

$$= \prod_g \left( \sum_q p(s|q, g) \cdot p(q|g) \right), \qquad (13)$$

$$= \prod_g \left( \sum_q \left( \sum_t p(t|q, g) \cdot p(s|t, q, g) \right) \cdot p(q|g) \right), \qquad (14)$$

$$= \prod_g \left( \sum_q \left( \sum_t p(t|q, g) \cdot p(s|t, q) \right) \cdot p(q|g) \right). \qquad (15)$$

One can notice that, when #$q = 1$ (i.e., only a single query exists), both variants converge to $\prod_g p(s|q, g)$, which can be written as Eq. 5. The modeling of each term in the above equations will be later demonstrated.

## 3.3 Ranking with Static and Stochastic Policies

We now discuss how to compute the search success given an intent and query, *i.e.*, $p(s|t, q)$, which can be defined as the success that at least one of the items is successful [1]:

$$p(s|t, q) = 1 - \prod_d^{\mathcal{D}} \left( 1 - p(s_d|t, q) \right). \qquad (16)$$

To compute the success of the item $d$ given an intent $t$ and query $q$, i.e., $p(s_d|t, q)$, either static ranking policy or stochastic ranking policy [9] can be applied.

***Static ranking policy.*** Let $r_d$ represent the event that $d$ is relevant and $\epsilon_d$ that $d$ is exposed to a searcher. We further define $s_d$ as the event that a search is successful via item $d \in \mathcal{D}$—i.e., $d$ is relevant and viewed by the searcher. In the absence of personalization and any difference across groups in how they inspect the retrieved results, we have:

$$p(s_d|t, q) = p(r_d|t) \cdot p(\epsilon_d|\sigma_q, \mu). \qquad (17)$$

Here $\sigma_q$ is a rank list of items given query $q$ produced by some model. $\mu$ is a user browsing model, which provides a mechanism to estimate the probability of exposure of an item $d$ in a retrieved ranked list of items $\sigma_q$ with respects to input query $q$. For example, the user browsing model behind the rank-biased precision (RBP) metric [19] assumes that the probability of the exposure event $\epsilon_d$ for $d$ depends only on its rank $\rho_{\sigma_q}$ in $\sigma_q$ and falls off exponentially further down the ranked list.

$$p(\epsilon_d|\sigma_q, \mu_{RBP}) = \gamma^{\rho_{\sigma_q}-1}, \qquad (18)$$

where the $\gamma$ is the patience factor and controls how deep in the ranking the searcher is likely to browse, "-1" is to force the position starting from zero. In this work, we use the RBP user browsing model, although alternative models could also be applied.

***Stochastic ranking policy.*** Diaz et al. [9] define a stochastic ranking policy $\pi$ as a probability distribution over all permutations of items in the collection. If the search system under inspection employs a stochastic ranking policy, then we can rewrite Eq. 17 as:

$$p(s_d|t, q) = p(r_d|t) \cdot \sum_{\sigma \sim \pi_q} p(\sigma|q) \cdot p(\epsilon_d|\sigma, \mu). \qquad (19)$$

## 4 Metric Comparison

One related metric to ours is the diversity concept proposed by Agrawal et al. [1], who frame their diversity objective as to maximize the probability of the searcher finding at least one relevant result given a query $q$ and a distribution over intents $t$. We rename their objective as Diversity-aware Search Success (DA-SS) defined as below:

$$p(\mathrm{S}^{\mathrm{DA}}|q) = \sum_t^\tau p(t|q) \cdot p(s|t, q), \qquad (20)$$

$$= \sum_t^\tau p(t|q) \cdot \left(1 - \prod_d^{\mathcal{D}} \left(1 - p(s_d|t, q)\right)\right). \qquad (21)$$

It can be noted that DA-SS is analagous to GA-SS as defined in Eq. 3, except it does not account for the variance in the likelihood of intents associated with a query across different demographic groups. However, optimizing the system towards diversity does not guarantee that group-aware success also improves. We omit this comparision here since relevant examples have already been illustrated in Section 1 (Case 2, Figure 1).

In this section, we use an additional toy example to study the relation among within-query GA-SS metric and its two across-query metric variants. We aim to highlight that improving one metric does not necessarily benefit others, and sometimes may show negative impacts. For simplicity, in the toy example, we assume all the items

retrieved are relevant to the corresponding intent and observed by searchers. Thus, for all those considered intents, the search success given an intent and query (i.e., Eq. 16) always equals one.

We focus on the toy example shown in Figure 2. Imaging there are two queries $q_1$ and $q_2$ in the full system with equal sampling probability, where each query is equally relevant to two intents $t_1$ and $t_2$. There are two equal-size user groups, where group $g_A$ is always interested in $t_1$ and group $g_B$ is always interested in $t_2$. Now, we consider nine different search systems, where each system retrieves one or two intents for each query. We then compare the GA-SS within each query and across queries by computing the corresponding search success. Notice that there are two ways to compute the overall GA-SS across queries due to different order of aggregation over the queries and user groups. As shown in Figure 2 where each row is a different retrieval result of a search system, we can clearly observe that the value change trend of each metric (column) is not exactly the same. For instance, compare two search systems where the first one retrieves $t_2$ for $q_1$ and $t_1$ for $q_2$, while the second one retrieves $t_2$ for $q_1$ and $t_2$ for $q_2$. Their GA-SS values within both queries are equal, while the overall GA-SS values across queries are not exactly the same when comparing $p(\mathrm{S}^{\mathrm{GA}}_{\Pi \Sigma})$.

## 5 Experiment and Analysis

Previously, we used some simple example to demonstrate that the different metrics are not always aligned; improving one metric might negatively affect others. In this section, we aim to conduct analysis on real-world datasets to further investigate the relation among these metrics with different ranking policies. Specifically, we aim to answer the following two research questions (RQs):

- **RQ1**: What is the impact of stochasticity on our proposed GA-SS for both within query and across queries scenarios?
- **RQ2**: What is the correlation between GA-SS and DA-SS in a single-query scenario, as well as the two variants of GA-SS across queries?

### 5.1 Task and Dataset

We conduct our analysis on the following two tasks in this work. We formulate both tasks into search problems.

***Query Auto-completion.*** Query auto-completion (QAC) is one of the most prominent features of modern search engines. The list of query candidates is generated according to the prefix entered by the user in the search box and is updated on each new key stroke [30]. Following each new character entered in the query box, search engines filter suggestions that match the updated prefix, and suggest the top-ranked candidates to the user. We use the Sogou query logs 2008 dataset[1] for this task. The queries, all in Simplified Chinese, were extracted from a Chinese search engine and include anonymized user IDs and click data. They span from 2008-06-01 to 2008-06-30, encompassing over 25 million typed queries. Under our setup, we define the key notations as follows:

- query $q$: all possible search prefixes;
- item $d$: the entire complete query;
- intent $t$: query clusters based on embeddings or urls;

---

[1]http://www.sogou.com/labs/resource/

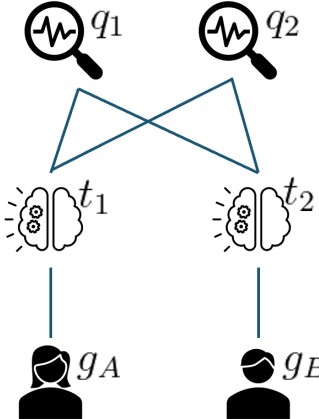

| Retrieved For $q_1$ | Retrieved For $q_2$ | GA-SS within Query $p(\text{S}^{\text{GA}}\|q_1)$ | $p(\text{S}^{\text{GA}}\|q_2)$ | GA-SS across Queries $p(\text{S}^{\text{GA}}_{\Sigma\Pi})$ | $p(\text{S}^{\text{GA}}_{\Pi\Sigma})$ |
|---|---|---|---|---|---|
| $t_1$ | $t_1$ | 0.5 | 0.5 | $\frac{1\times0+1\times0}{2}=0.0$ | $\frac{1+1}{2}\times\frac{0+0}{2}=0.0$ |
| $t_1$ | $t_2$ | 0.5 | 0.5 | $\frac{1\times0+0\times1}{2}=0.0$ | $\frac{1+0}{2}\times\frac{0+1}{2}=0.25$ |
| $t_1$ | $t_1+t_2$ | 0.5 | 1.0 | $\frac{1\times0+1\times1}{2}=0.5$ | $\frac{1+1}{2}\times\frac{0+1}{2}=0.5$ |
| $t_2$ | $t_1$ | 0.5 | 0.5 | $\frac{0\times1+1\times0}{2}=0.0$ | $\frac{0+1}{2}\times\frac{1+0}{2}=0.25$ |
| $t_2$ | $t_2$ | 0.5 | 0.5 | $\frac{0\times1+0\times1}{2}=0.0$ | $\frac{0+0}{2}\times\frac{1+1}{2}=0.0$ |
| $t_2$ | $t_1+t_2$ | 0.5 | 1.0 | $\frac{0\times1+1\times1}{2}=0.5$ | $\frac{0+1}{2}\times\frac{1+1}{2}=0.5$ |
| $t_1+t_2$ | $t_1$ | 1.0 | 0.5 | $\frac{1\times1+1\times0}{2}=0.5$ | $\frac{1+1}{2}\times\frac{1+0}{2}=0.5$ |
| $t_1+t_2$ | $t_2$ | 1.0 | 0.5 | $\frac{1\times1+0\times1}{2}=0.5$ | $\frac{1+0}{2}\times\frac{1+1}{2}=0.5$ |
| $t_1+t_2$ | $t_1+t_2$ | 1.0 | 1.0 | $\frac{1\times1+1\times1}{2}=1.0$ | $\frac{1+1}{2}\times\frac{1+1}{2}=1.0$ |

Figure 2: A toy example for the GA-SS metric comparison. Two queries $q_1$ and $q_2$ have equal sampling probability, where each query is equally relevant to two intents $t_1$ and $t_2$. Two searcher groups are of equal size, where group $g_A$ is always interested in $t_1$ and group $g_B$ is always interested in $t_2$. In practice, a small positive value should be added on the success for a smoothing to avoid zero. We ignore it in this toy example for simplicity. We observe that the patterns of change across each metric variant do not consistently align, which suggests that each metric variant captures different aspects of search success.

- user group $g$: classified into active group and inactive group based on the interaction frequency (due to lack of user demographic information).

***Movie Recommendation.*** Movie recommendation is a widely studies task in the field of information retrieval and has been widely applied in real-world systems. We use the MovieLens 1M dataset[2], which is the largest version of MovieLens dataset that contains user demographic information. The dataset contains 6,040 users and 3,706 items with 1 million user-item interactions. We adapt this dataset to a search task and define the key notations as follows:

- query $q$: director (information gathered from IMDB[3]);
- item $d$: the recommended movie;
- intent $t$: different movie genres;
- user group $g$: female group and male group as denoted in the dataset.[4]

## 5.2 Experiment Setup

To compute the entire metric value, we first discuss how to compute each decomposed terms iof GA-SS, especially referring to Eq. 10 and Eq. 15. Without loss of generality, we use the *query auto-completion* task as an example in the following narrative, where $q$ refers to search prefix, $d$ refers to the entire complete query, and $t$ refers to query clusters based on embeddings (or urls).

- $p(t|q,g)$: The notation can be further decomposed as below:

$$p(t|q,g) = \sum_d p(t|d,q,g) \cdot p(d|q,g), \qquad (22)$$

$$= \sum_d p(t|d) \cdot p(d|q,g). \qquad (23)$$

[2]https://grouplens.org/datasets/movielens/1m/

[3]https://www.imdb.com/

[4]Note that gender is treated as a binary class due to the available labels in the dataset. We do not intend to suggest that gender identities are binary, nor support any such assertion.

Both of the two terms in the final equation can be derived from the original data. $p(t|d)$ denotes the relateness of an intent given an entire query, which can be computed based on the embeddings of intents and queries. These embeddings can be obtained from some pretrained text encoder, such as BERT [8]. $p(d|q,g)$ is the probability that $d$ is the query the searcher from group $g$ submits for the input prefix $q$, can be directly estimated based on the original data.

- $p(q)$: The probability of a prefix $q$ being launched by any arbitrary searcher can be measured based on frequency on the original data.
- $p(q|g)$: The probability of a prefix $q$ being launched by searchers from group $g$ can be measured based on frequency on the original data.
- $p(s|t,q)$: As discussed in Section 3.3, $p(s|t,q)$ can be further decomposed using either static or stochastic ranking policies. Referring to Eq. 16, Eq. 17, and Eq. 19, we now need to compute three terms at most: $p(r_d|t)$, $p(\epsilon_d|\sigma_q,\mu)$, and $p(\sigma|q)$. For $p(r_d|t)$, it represents the relatedness of entire query $d$ with respect to some given prefix $t$, which can be computed through the similarity of the corresponding embeddings. $p(\epsilon_d|\sigma_q,\mu)$ estimates the probability of exposure of output $d$ based on some user browsing model $\mu$ applied on top of a rank list $\sigma$ produced by some model that we want to evaluate given input $q$, which can be computed through Eq. 18. $p(\sigma|q)$ is sampled from some arbitrary policy $\pi$.

## 5.3 Method

We are interested in son evaluating stochastic ranking policies for their effectiveness in distributing exposure among items. To facilitate this, we initially describe a technique for creating stochastic ranking policies using a static ranking model. By taking a static ranker with its associated relevance scores for items relative to a query $q$, we apply the Plackett-Luce (PL) model [17, 24] to generate

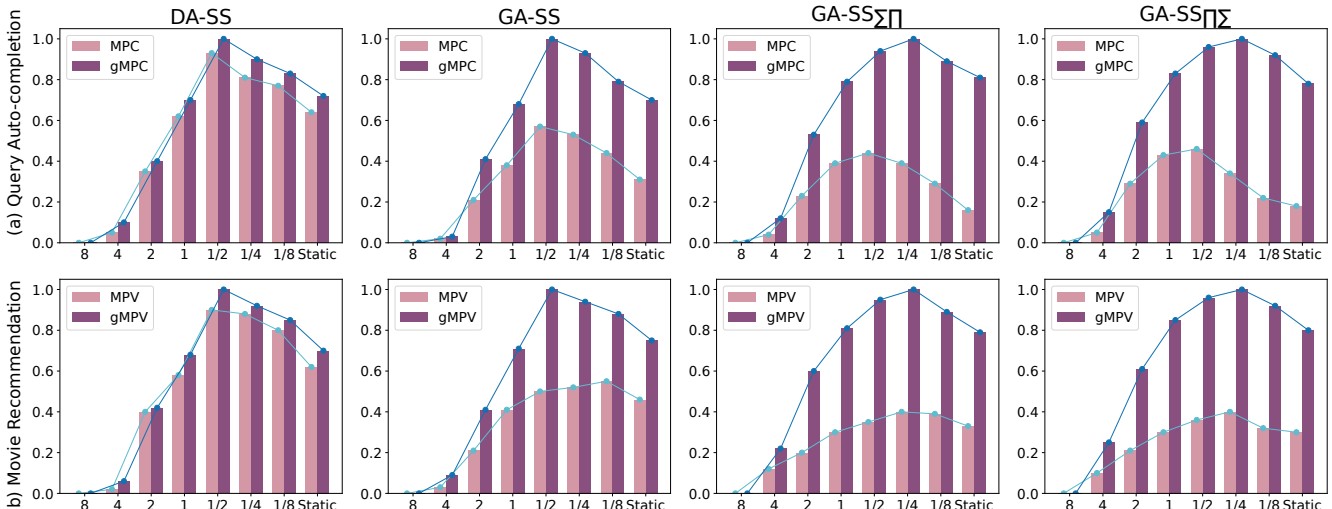

Figure 3: Behavior of different metrics for a stochastic ranking policy—generated by randomizing the MPC/MPV and gMPC/gMPV models using Plackett-Luce. The first row shows the impact of different stochasticity the query auto-completion, while the second row shows the result on movie recommendation. For consistency, we normalize each of the metric values between 0 and 1 using min-max normalization in each subfigure. The x-axis shows the values of $\beta$, where a larger value indicates more randomization.

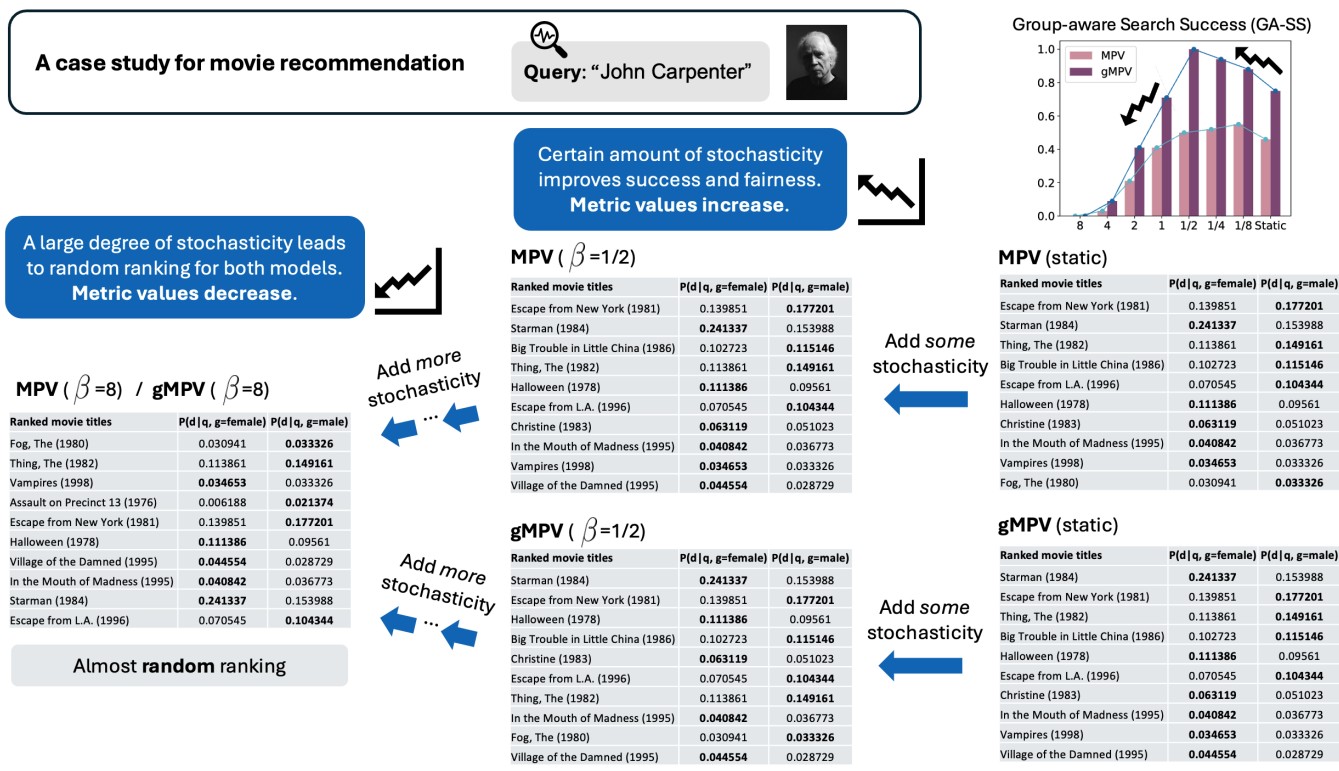

Figure 4: A case study on impact of stochasticity on different ranking models. We use the movie recommendation as an example and report the top-10 ranked movies with respect to the query (director): *"John Carpenter"*. As shown above, when adding some moderate amount of stochasticity to the ranking model, the success and fairness both improve thus leading to increased metric values (e.g., GA-SS). However, when a large amount of stochasticity being added, rankings from both models converge to a random ranking, leading to decreased metric values.

multiple rankings. The PL model adheres to Luce's axiom of choice, which asserts that the probability of selecting an item over another is independent of the other items present [16, 17]. Specifically, the PL model constructs a ranking by iteratively sampling items without replacement from the collection with probability distribution $p(d|q)$ defined as below:

$$p(d|q) = \frac{\exp(r_{d,q}/\beta)}{\sum_{d' \in \mathcal{D}} \exp(r_{d',q}/\beta)}, \quad (24)$$

where $r_{d,q}$ is the relevance score estimated by the static ranker for item $d$ with respect to query $q$. The parameter $\beta$ is the softmax temperature. A larger $\beta$ corresponds to more stochasticity in the ranking. For example, when $\beta = 8$, the probability distribution over all permutations is almost uniform and the stochastic policy approaches a fully random ranking model. As a corollary, when $\beta$ decreases the stochastic policy converges to the static ranking policy, which is a ranking of items sorted by their estimated relevance score $r_{d,q}$ in descending order for each query $q$.

For the query auto-completion, we generate stochastic ranking policies by applying this post-processing step, with different values of $\beta$, over two models: (i) *Most Popular Completion (MPC)*, one of the most widely used baselines that can be regarded as ranking based on $p(d|q)$, and (ii) *Group-aware Most Popular Completion (gMPC)*, proposed by ourselves, which ranks items based on $\prod_g p(d|q, g)$. For the movie recommendation, we use the same models but rename them as (i) *Most Popularly Viewed (MPV)* and (ii) *group-aware Most Popularly Viewed (gMPV)* for distinction. For both tasks, we sample 100 rankings for each query during evaluation. We employ the RBP user browsing model and set the patience factor $\gamma = 0.8$. We select different values for $\beta$ in the range of 1/8 to 8 for introducing different degree of stochasticity in our ranking, and compare with the deterministic ranking policy.

## 5.4 Impact of Stochasticity (RQ1)

To investigate the impact of stochasticity, we first visualize how different values of $\beta$ influence different metrics. Our analysis is based on stochastic ranking policies that use the MPC/MPV and gMPC/gMPV models as the underlying static ranking models. We report the metric values on averaged DA-SS within query, averaged GA-SS within query, and two variants of GA-SS across queries based on different aggregation strategies, as shown in Figure 3. For consistency, we normalize each of the metric values between 0 and 1 using min-max normalization.

Our first observation is that MPC/MPV show similar performance to the corresponding group-aware version method on DA-SS, while largely inferior to the counterparts on other three metrics. This is unsurprising since DA-SS metric is group-unaware thus models that do not incorporate group information can still achieve a high score, while all other metrics are group-aware requiring the model to take group attributes into consideration for a good performance. Our second observation is that as $\beta$ increases, the values on all metrics first increase and then decrease towards zero. This pattern aligns with expectations given that a larger $\beta$ corresponds to a more random ranking policy, where the original static relevance estimates have a smaller influence, which consequently results in low search success. Interestingly, the metric values first increase and then decrease, illustrating a trade-off between success

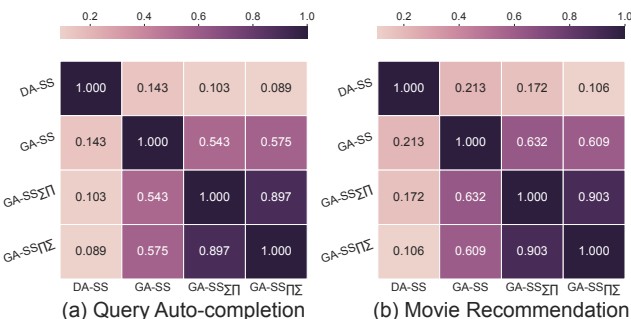

Figure 5: The Kendall rank correlation between different metrics on the two tasks and datasets we studied.

and fairness within our metric. This suggests that a certain level of randomness can optimize this trade-off. As illustrated in Figure 3, the degree of optimal stochasticity varies across different metrics and datasets.

***Case Studies***. To visualize the impact of stochasticity more clearly, we use the movie recommendation as an example and we investigate the effect of stochasticity on ranking models, specifically focusing on the output for the director query *"John Carpenter"*. Figure 4 shows that incorporating a moderate level of stochasticity into the ranking process maintains high success (i.e., highly satisfied movies ranked higher) and improves fairness to different demographic groups (i.e., highly satisfied movies for female group and male group both occur in the top of the list), as reflected by improved values in metrics such as Group-aware Search Success (GA-SS). Conversely, introducing excessive stochasticity causes the rankings provided by both models to approach a uniform policy, which in turn leads to a decline in the performance metrics. The top-10 movie rankings are used to illustrate these effects across different degrees of stochasticity in the models.

## 5.5 Correlation Analysis (RQ2)

Next, we show the cross-metric analysis to understand the correlation of different search success metrics. To study this, we use the same ranking models, with seven different levels of stochasticity in each case (i.e., $\beta = 8, 4, 2, 1, 1/2, 1/4, 1/8$). For each metric, this gives us $2 \times 7 = 14$ combinations of model and stochasticity level. Now for every pair of search success metrics, we compute the Kendall rank correlation [13] to quantify the agreement between the two metrics with respect to the ordering of the ranking model instances. For the two metrics measured within a single query, DA-SS and GA-SS, we compute a single value for them by averaging over all queries. We perform the analysis on the query auto-completion and movie recommendation, respectively.

As shown in Figure 5, we observe that the correlation between the DA-SS metric and the other four search success measures is typically low, which is expected since DA-SS is the only group-unaware metric. GA-SS metric (within query) shows stronger correlation with the two variants of GA-SS across query metrics. This is also aligned with our expectations due to their similar metric formulation and consideration on group attributes but focusing on different

query aggregations. The two variants of GA-SS across query metrics, GA-SS$_{\sum \Pi}$ and GA-SS$_{\Pi \sum}$ show the strongest correlation and their only difference lie in the different order for computing the metrics over queries and groups. Overall, the correlation matrices align with the behavior we would expect by comparing Eq. 5, Eq. 10, Eq. 15, Eq. 21, and are consistent over two datasets.

## 6 Search Success and Relevance Metrics

Our group-aware search success metric also shows connections to rank-weighted relevance metrics in search, such as NDCG, which are generally used for measuring quality of service. The search success can be decomposed into two components (see Eq. 17): the probability that a searcher observes the result and the probability that the result leads to a success. Analogously, rank-weighted relevance metrics can also be decomposed into two components (see Eq.(3.7) in [34]): the probability that a searcher observes the result and the expected reward/score from the result. Notably, the first component is identical and depends on some user browsing model.

Thus, our proposal inspires alternative and various definitions of quality of service in search. By using relevance metrics like NDCG, one can first compute a score for each user per query. Then the quality of service of search can be defined by adopting a two-level aggregation approach similar to sum-of-product and product-of-sum we designed for $p(\text{S}_{\Pi \sum}^{\text{GA}})$ and $p(\text{S}_{\sum \Pi}^{\text{GA}})$: either aggregating results first across groups and then across queries, or the reverse. In the aggregation process, both arithmetic and geometric means are viable, each emphasizing different attributes. The arithmetic mean focuses on overall user satisfaction by weighting each score equally, while the geometric mean offers a more balanced perspective by reducing the influence of outliers. While a thorough analysis of these metrics is outside the scope of this paper, they are deserving of further investigation in future research.

## 7 Conclusion

In this paper, we have addressed the limitations of traditional measures of search success that often neglect the diverse information needs across different searcher demographic groups. To address this, we introduced a new metric called Group-aware Search Success (GA-SS), which redefines search success to ensure success across all demographic groups. We developed a detailed mathematical framework to calculate GA-SS, using both static and stochastic ranking policies, and allowing to incorporate any user browsing model. Furthermore, we proposed the Group-aware Most Popular Completion (gMPC) ranking model, designed to better accommodate demographic variations in user intent, thereby aligning more closely with the diverse needs of all user groups. We empirically validated our metric and approach using two real-world datasets. These studies illuminate the effects of stochasticity and the intricate relationships among various search success metrics. Our results underscore the importance of adopting a more inclusive approach to measuring search success and inspire future investigations into the quality of service based on relevance metrics.

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
