# OpenReview forum: "Towards Group-aware Search Success"
_ACM.org/SIGIR/ICTIR/2024/Conference — ICTIR 2024_

### Official Review · Reviewer_oBgC · 2024-05-19

**Rating:** 0
**Confidence:** 4

**Objective Part Of Review:**

The paper presents metrics for measuring group-aware search success, and applies these in two settings (query autocompletion and movie recommendation).

The related work seems quite complete, but the novelty of the authors' contribution is not that well situated in all this related work. Especially, in "stochastic ranking policy" it is clear that the authors build on [9] (Diaz et al) but it is not clear what they add, and how the work of [9] is either integrated, build upon, or modified. Also, at the end of 2.2 the authors write that "our concept concerns item exposure" but it was my understanding that this is the case in most approaches to evaluating fairness - that should be stated more clearly.

Given the paper's main objective, I do not fully understand why the authors did not relate their work to the research carried out in context of the TREC fairness track. Why are the datasets developed at this track of TREC not applicable to your work? I'm not an expert on fairness, but I have been surprised to not see a mention of this line of work, nor a justification why the corpora used there are not considered for this paper.

Finally, I was wondering what the intended role of the experiments has been for this work. What should the experiments show? This is not explicitly stated up-front. For me, it remained unclear how the experiments presented help us understand the problem setting of group-related preferences that should be accommodated somehow.

**Subjective Part Of Review:**

Regarding the evaluation that is presented, I am wondering whether we should expect movie preference to be influenced by gender, and whether we would want to design systems that would, say, only recommended romcoms to women and scifi to men - what is the rationale here? Maybe the authors should elaborate a bit on the ethical aspects of their envisioned approach to evaluation / system design.
(I'm not claiming gender should never be a consideration in movie taste, but just deriving the evaluation of a movie recommender from historic data does run the risk of perpetuating biases that should not have existed. What do the authors think about this?)

---

### Official Review · Reviewer_m5g4 · 2024-05-21

**Rating:** 1
**Confidence:** 3

**Objective Part Of Review:**

Authors claim that the traditional measures of search success often fail to consider the diverse information needs of different demographic groups. To address this, they introduce a new metric called Group-aware Search Success (GA-SS), which redefines search success to ensure satisfaction across all demographic groups. GA-SS is computed using a  mathematical framework that incorporates both static and stochastic ranking policies, along with user browsing models for more accurate assessments. They also propose the Group-aware Most Popular Completion (gMPC) ranking model to better account for demographic variations in user intent. The metric and approach are empirically validated with two real-world datasets—one on query auto-completion and the other on movie recommendations. The results demonstrate the significance of stochasticity and the complex interactions among various search success metrics. The findings advocate for a more inclusive approach to measuring search success and inspire future research into improving search service quality.

This is an interesting research.

Authors used MovieLens and query auto completion datasets to evaluate

**Subjective Part Of Review:**

the mathematical notations need to be explained; for example, equation 3 that depicts the relationship between s and g

This paper is interesting as it addresses a specific problem of evaluation.

---

### Official Review · Reviewer_mGyZ · 2024-05-23

**Rating:** -1
**Confidence:** 3

**Objective Part Of Review:**

This paper proposes a Group-Aware Search Success (GA-SS) metric to better evaluate the search result's quality across different demographic groups. The proposed metric is well-motivated and carefully-designed. My main concern is that the application of this metric is limited since its computation involves user intents. There also exist some problems in experiments. I suggest a weak reject for this paper.


Strengths:
1. The basic assumption that **search success is being achieved _iff_ all demographic groups find success in their search outcomes** is reasonable and well-motivated. The definition and derivation of the proposed metric is solid.
2. One toy example and studies on two real-world datasets confirm the validation of the proposed metric. In particular, the correlation analysis across different success metrics in Section 5.5 shows that GA-SS is different from DA-SS.

Weaknesses:
1. The calculation of the proposed metric GA-SS involves the search intents $t$. However, it is difficult (if not impossible) to obtain the search intents of users and groups under real situations and in most datasets. Therefore, the application of the proposed metrics is relatively limited.
2. Some calculation in Figure 2. is confusing. For example, when the system retrieves $t_1$ for $q_1$ and $t_2$ for $q_2$, GA-SS within query should both be 0 for $q_1$ and $q_2$ since $g_B$ is not satisfied under $q_1$ and $g_A$ is not satisfied under $q_2$. (Indeed, I think all 0.5  under the GA-SS within query metric should be 0.) The authors need to check this calculation and should better demonstrate the detailed calculation process.
3. Section 3.3 is somehow irrelevant to the topic of this paper.

Minor problems:
Some references in Section 2.1 (diversification) are missing.

**Subjective Part Of Review:**

Overall, this paper is well-written and relevant to this proceeding. The methods are original. The proposed metric may inspire future studies in the ICTIR community.

---

### Meta-Review · Area_Chair_qLSG · 2024-05-29

**Recommendation:** Accept (Oral)
**Confidence:** 4

**Metareview:**

The observation at the basis of this paper is that  traditional measures of search success are often overlooking the varying information needs of different demographic groups, which is indeed very interesting.
To address this issue a new metric is proposed to reshape the notion of serachj success, with the aim of ensuring that all demographic groups achieve satisfaction from search outcomes. Although the paper addresses an interesting issue, it has some limitations, e.g., the formalization has some unclear aspects, as well as some unclear aspects would need clarification in relation to evaluations.